# Effect of Cold Pressing Deformation on Microstructure and Residual Stress of 7050 Aluminum Alloy Die Forgings

**DOI:** 10.3390/ma16145129

**Published:** 2023-07-20

**Authors:** Huiqu Li, Liang Wang, Weiwei He, Liqiang Cheng, Junzhou Chen, Linna Yi

**Affiliations:** 1AECC Beijing Institute of Aeronautical Materials, Beijing 100095, China; leehuiqu@163.com (H.L.); liangwang0010@163.com (L.W.); 19385072@hotmail.com (W.H.); chengliqiang1215@163.com (L.C.); junzhouchen@126.com (J.C.); 2Beijing Engineering Research Center of Advanced Aluminum Alloys and Applications, Beijing 100095, China

**Keywords:** 7050 aluminum, die forging, cold compression, residual stress

## Abstract

Large-scale, high-strength aluminum alloy forgings are essential components in the aerospace industry, with benefits including increasing strength and decreasing weight. Accurate shape-property control is the secret to forging quality. This study uses the alloy 7050 to experimentally evaluate the parametric influence of cold compression on residual stress and mechanical characteristics. The evolutions of mechanical properties, microstructure and residual stress are theoretically studied using various cold compression strains from 1% to 5% on an equivalent part, of which the results are further applied on a complicated rib-structured die forging. It is demonstrated that increasing the compression strain reduces the tensile strength of the material, but has little impact on conductivity and fracture toughness. According to the TEM results, compression also encourages the precipitation and growth of precipitated phases, particularly in positions with high dislocation densities after aging. Cold compression significantly reduces residual stress; nevertheless, as compression strain increases, residual stress first decreases and then increases. With the use of rib-structured forging, it is observed that the compression strain for 7050 aluminum alloy ranges from 2% to 4%, and the combined pressing method of the rib and web improves the uniformity of residual stress.

## 1. Introduction

Al-Zn-Mg-Cu high-strength aluminum alloy is the preferred material for structural parts in the aerospace field due to its excellent comprehensive properties such as low density, high strength, and good processing performance [1,2]. With a rising number of massive forging machines, large-scale aluminum alloy forgings are widely being designed and used to enhance structural integrity and decrease aircraft weight. Due to its exceptional comprehensive qualities, including strength, fatigue performance, and hardenability, the 7050 aluminum alloy is frequently used for aeronautical die forging [3,4,5,6,7]. Due to the unequal shrinking of internal and external sides brought on by an abrupt temperature difference during the quenching process, residual stresses are inexorably generated during the manufacturing process in aluminum alloy forgings. The further release of the residual stresses during the machining process causes unwanted deformation, such as warping and twisting, which reduces the geometric correctness of the parts and has an impact on the assembly process. Therefore, reducing residual stress during die forging production is crucial, particularly when producing large quantities of vital structural elements for the aircraft industry.

The measurement and control of residual stress have been widely investigated. Pan et al. [8] studied the influence of quench sensitivity on residual stress. Zhang et al. [9] investigated the geometric error after cold compression using the stress contour method. Robinson et al. [10] investigated the residual stress distributions using neutron diffraction and deep hole drilling. Gao et al. [11] proposed a thermal-vibratory stress relief method to decrease residual stress. Liu et al. [12] applied a cladding quenching method to eliminate the quenching residual stress of the 7085 Al alloy. The most efficient method of removing residual stress is cold deformation immediately following quenching, in which the internal residual stress is transferred by utilizing the planned plastic deformation strain brought on by cold deformation. Homogeneous cold deformation for large-scale die forging production is frequently constrained by changes in the position of structural features, such as rib-structured parts. After determining the proper cold deformation strain range, it is crucial to design a specific deformation strain based on the part geometry for complicated parts. Yao et al. [13,14,15] studied the evolution of residual stress during segmented cold pressing in 7050 and 7085 aluminum alloy forgings using a finite element simulation, and a 1~3.5% segmented cold pressing can reduce the residual stress of aluminum alloy forgings by 70~90%. Wang et al. [16] established a theoretical model to predict and control machining deformation caused by residual stress. Tang et al. [17] proposed an integrated physically based model to predict the mechanical properties in the hot forging of aluminum alloys. Jiang et al. [18,19] revealed the kinetics of dynamic and static softening during the forging process. These researchers analyzed the influence of cold compression on residual stress through experiments and simulations. However, the evolution of the microstructure and mechanical properties during cold compression is rarely reported, resulting in the obstacle of the material property control of complex parts during residual stress reduction.

Due to the development of large-scale forging equipment, integrated cold pressing for large-scale die forging products has attracted increasing attention in an effort to shorten the process, streamline the process flow, and increase production efficiency. Wu et al. simulated the solid-solution, quenching, and cold pressing process of a long-shaft rib plate die forging made with the 7050 aluminum alloy, and predicted the distribution of residual stress after the quenching of the forgings. They found a 3% cold deformation strain can reduce residual stress significantly [20]. Zhai et al. [21] predicted the residual stress distribution of long rib-free forgings with dimensions of 1500 mm × 500 mm × 200 mm after quenching using thermodynamic calculation software and plastic forming software, and found a 3% cold pressing deformation and 200 mm feed amount can eliminate the quenching residual stress. However, the optimized cold compression strain for both mechanical properties and residual stress control is still unknown.

The limited investigations of the interaction effect between residual stress and mechanical properties affects the quality of complex forging products during the cold compression process of complex forging products. This study uses a large-scale rib-structural forging and an analogous sample to evaluate cold compression experimentally for the integrated control of forging mechanical properties and residual stress. Using an equivalent sample, it is first determined how cold compression strains affect mechanical characteristics and residual stress. The microstructural analysis is presented to show how characteristics change over time. The impacts of the cold compression method are further investigated using a large-scale rib-structural forging based on the findings of an equivalent sample. The suggested approach of optimum cold compression takes into account mechanical qualities, residual stress, and geometric precision. The findings provide insight into the industrial use of cold compression on intricate structural components.

## 2. Experimental Methods

The forging material was the 7050 aluminum alloy and the alloying elements were Zn (6.27 wt.%), Cu (2.03 wt.%), Mg (1.96 wt.%), and Zr (0.10 wt.%). In this work, an equivalent sample was selected to analyze the formation mechanism during cold compression, and a large-scale rib-structural forging was studied for the industrial application of die forging. The size of the forging stock of an equivalent part was 200 mm × 150 mm × 140 mm, while the die forging stock was designed as shown in Figure 1 to have a size of 1316 mm × 406 mm × 140 mm. The residual stresses were caused by uneven cold shrinkage inside and outside during quenching. Therefore, all stocks were solid solutions treated at 477 °C for 210 min and were quenched in water at room temperature. Then, cold compression was applied to the samples, where the compression strains of the equivalent samples were 1%, 2%, 3%, 4%, and 5%, and those of the die forging stock were optimized to be 2~3%. For the die forging stock, cold compression was carried out by using a special cold compression mold, and according to the structural characteristics of the die forgings, cold compression was applied to eliminate residual stress by rib and web pressing, as shown in Figure 2. For the process integrity of the forging production, a two-stage aging process was applied where the samples were treated at 121 °C for 6 h, and then heated to 177 °C and held for 8 h, as list in Table 1.

The tensile samples were cut from the center of the material, and the tension direction was along the L direction, as shown in Figure 1. Three samples of room temperature tension were taken and repeated for result accuracy; the samples had a gauge length of 32 mm and a diameter of 6 mm, and their size followed the standard of ASTM B557 [22]. The fracture toughness was tested according to the standard of ASTM E399 [23], and the samples were cut from parts along the L-T and S-L directions with a size of 20 mm × 46 mm × 50 mm. The conductivity results were tested on the surface of the tensile samples and repeated three times following the standard of ASTM E1004 [24]. Transmission electron microscopy (TEM) testing was performed for samples of various compression strains using a Philips CM-12 machine (Philips, Holland), where specimens were prepared under the conditions of −20~30 °C and 15~20 V in an electrolytic double-spray thinning machine.

The residual stresses of the equivalent parts and rib-structural forgings were detected by the ultrasonic method and blind-hole method. The Masterscan-380M ultrasonic flaw detector produced by Sonatest (Milton Keynes, UK) was used for ultrasonic testing with a center frequency of 5 MHz and a wafer diameter of Φ 12.7 mm. The ultrasonic method is a non-destructive analysis method for determining residual stress based on the self-balancing characteristics of residual stress. During ultrasonic residual stress testing, the sound velocity was changed with the material residual stress, and the residual stress was qualitatively obtained according to the change in velocity, where the increase in the difference in sound velocity denotes a larger residual stress [25]. The blind-hole method was tested using the ASMB2-32 (Jinan Sigmar, Jinan, China) residual stress detector, and seven positions were selected for the die forgings.

## 3. Stock Results

### 3.1. Mechanical Properties of Stocks

Figure 3 displays the conductivity and room temperature tensile characteristics of various cold compression strains. The tensile strength reduces by about 30 MPa and yield strength decreases by about 50 MPa when the cold compression amount gradually increases from 1% to 5%. The elongation of forgings gradually increases. With the increase in the cold compression strain, the conductivity of forgings changes little, implying the aging process meets the requirements. Figure 4 shows the fracture toughness test results. The fracture toughness changes little with the increase in cold compression, and the change value is within 5 MPa·m^1/2^.

### 3.2. Residual Stress of Stocks

Figure 5 depicts the residual stress distribution cloud of various cold compression stresses using ultrasound. The difference in sound velocity is indicated by a color difference, and the residual stress increases with the difference. By increasing the compression strain, the residual stress is markedly reduced. Figure 5a depicts the specimen’s cloud prior to cold compression; the significant velocity difference between the surface and core denotes the specimen’s significant residual stress difference. During the quenching process, the outside of the material shrinks faster with the faster cooling rate than the inside of the material, which is attributed to the temperature gradient from the surface to the core. When the temperature of the outside is too low to shrink, the inside still has a higher temperature and continues to shrink over time but is restrained by the outside material, resulting in the state of tensile stress in the core and compressive stress at the surface [26]. Figure 5b–f shows specimens that have undergone cold compression. The sound velocity difference between the treated specimens and the control portions is noticeably lower, indicating that the residual stress has diminished.

The sound velocity dispersion coefficients of several samples are subsequently obtained to conduct a qualitative analysis of the residual stress. The sound velocity difference coefficients for various cold compression and aging processes are compared in Figure 6. The compression strain is increased while the residual stress is first lowered. After cold compression, the residual stress is further reduced by age while maintaining the variation trend. Before cold compression, the coefficient of the initial sample is 0.165; after 1% compression, it is 0.085. The coefficient drops to 0.062 at 3% and then increases to 0.072 when the compression increases from 1% to 5%. The coefficients are further reduced during the aging process, with the sample without cold compression falling to 0.148, while samples under 1%, 3%, and 5% cold compression are, respectively, 0.065, 0.052, and 0.062. As a result, 2% to 4% of compression strain is recommended.

### 3.3. Microstructure Morphology

Figure 7 shows the EBSD results of samples with various compression strains. It is shown that the samples after cold compression present a recrystallized crystal morphology. The recrystallized grain proportion of the sample with a compression strain of 1% is significantly higher than that at 5%. 

Figure 8 plots the TEM results of samples with various compression strains. Figure 8a,b illustrates the dislocation variation between different compression strains, where the dislocation is presented by the blue arrows. The sample with the 5% compression strain has a large dislocation density, and a dislocation cell structure is formed due to dislocation entanglement; meanwhile, the dislocation density of the sample with the 1% compression strain is little. Figure 8c–f presents the precipitation-phase morphology on the grain boundary of different materials, where the precipitation phases are presented by the black arrows. The grain boundary precipitation phase for the 1% strain sample is continuously distributed and its grain size is 50 µm. However, the precipitation-phase spacing is increased as the strain increases from 1% to 5% and shows intermittent distribution, and the average grain size is about 100 μm. Figure 8g,h shows the morphological distribution of the precipitated phase in the crystal, where the precipitation phases are presented by the yellow arrows. The precipitated phase in the crystal of the 1% strain sample is smaller than that at 5%. In summary, the increase in the cold compression strain increases the dislocation density and induces a larger proportion of the precipitated phase, and the former increases the material strength while the latter has an opposite effect. As the cold compression strain increases from 1% to 5%, the tensile strength is reduced by 30 MPa and the yield strength is decreased by 50 MPa, indicating that precipitation plays a greater role in the material properties.

## 4. Results of Die Forging

### 4.1. Mechanical Properties with Various Compression Methods

In Figure 9, the mechanical characteristics of forgings with rib structures were compared to those of similar parts. For rib-structural forgings, two cold compression techniques were used: rib-and-web combined compression and rib compression, where the rib’s cold compression strain is 2.2% and the web’s is 3.1%. It is demonstrated that the combined compression sample’s mechanical characteristics, including tensile strength, fracture toughness, and conductivity, are comparable to those of components with compression strains of 2% and 3%. Since the web of the rib compression sample is not plastically deformed, where the strength is reduced with compression strains, it has a somewhat higher tensile strength than other samples. 

### 4.2. Residual Stress of Different Cold Compression Methods

Figure 10 displays the findings of rib-structural forgings of unexpanded samples, rib-and-web combined compression, and rib compression using ultrasonic residual stress. The unpressed samples exhibit high residual stresses, with stresses varying depending on the position along the 40 mm thick web and the specific cooling direction used during the quenching procedure when the component was submerged in water. Figure 10b illustrates how, in forgings with rib compression, the material flows to the web during rib pressing but the web is left undisturbed. As a result, the metal at the root of the ribs is subjected to pressure in two directions and the stress increases, causing the difference between the connection area of the web and ribs and the central web to remain significant. Due to the simultaneous deformation of the ribs, webs, and connecting area, Figure 10c demonstrates that the residual stress distribution of die forgings with combined compression is relatively uniform.

The small-hole approach was used to investigate the residual stresses, and Figure 11 shows the positions that were discovered. Figure 12 depicts the residual stress values for various forging portions, with the values for the unpressed samples being much greater than those for the crushed sections at or above 110 MPa. By using the rib compression method, the residual stress is reduced at both the rib and web places. The average residual stress is 106.4 MPa, which is lower than the 148.4 MPa residual stress of the initial parts, having been reduced by 28.3%. The residual stress of the web is reduced by the rib-and-web combined compression method. In comparison to an unpressed sample, the average residual stress using the rib-and-web combined compression method is 53.3% lower at 69.2 MPa. According to indications, the combined compression approach using the rib and web is more effective in removing residual tension.

### 4.3. Die Forging Machining Result

The machining experiment was further carried out, and ten geometric deformation measurement points are shown in Figure 13. The geometric deformations of various positions are shown in Figure 14, where the unpressed sample exhibits a significant inaccuracy in the middle of the sections with positions of 2~4 and 7~9. The position 3 error is the spot with the largest error, measuring 4.7 mm. By using the cold compression method, the errors are reduced, indicating that the machining deformation is suppressed as residual stress decreases. Due to the lower residual stress, the combined compression technique’s errors are substantially lower than those of the rib compression method; the combined compression method’s maximum error is 0.3 mm, whereas the rib compression method’s is 0.49 mm. According to the aforementioned findings, rib and web compression at a compression strain of 2~3% is helpful in reducing residual stress and improving geometric precision while maintaining the material’s mechanical properties.

## 5. Conclusions

Forging on a large scale requires precise control of the residual material stress. This study used an equivalent part and aerospace rib-structural parts made of aluminum alloy 7050 to experimentally explore the effects of cold compression strain on mechanical characteristics, residual stress, and geometric error. In the meantime, the various compression techniques were described in depth to offer recommendations for parts with complex shapes. In addition to shedding insight into cold compression’s commercial uses for rib-structural elements, this research offers a recommended cold compression strain that optimizes both characteristics and residual stresses. The following are the key conclusions: (1)While fracture toughness and conductivity showed no change, the room temperature tensile strength decreased as the cold compression strain increased. The residual stress increased with an increase in the cold compression strain, with the optimal cold compression strain falling between 2% and 4%.(2)Increasing the cold compression strain boosts energy storage and material dislocation density, which encourages precipitation and the expansion of the precipitated phase during aging.(3)The rib-and-web combined compression method is more effective at reducing residual stress and at improving geometric correctness in rib compression. In the meantime, the homogeneity of mechanical characteristics on rib-structural sections can be improved by the combined compression of the rib and web.

## Figures and Tables

**Figure 1 materials-16-05129-f001:**
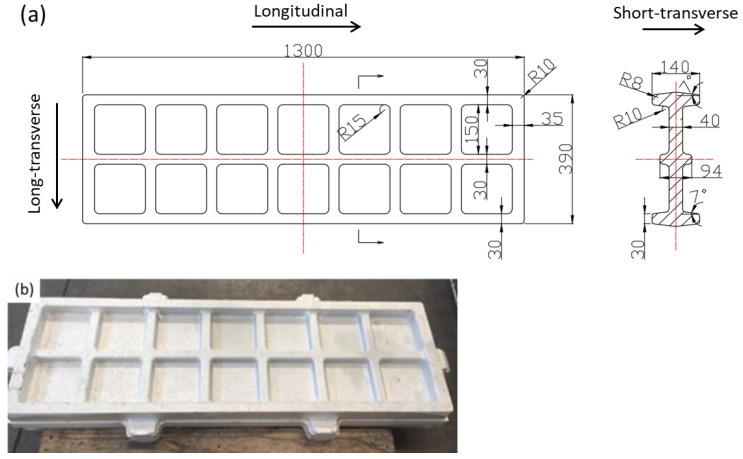
Die forging stock dimensions: (**a**) forging drawing (The red line indicates axis); (**b**) physical forging.

**Figure 2 materials-16-05129-f002:**
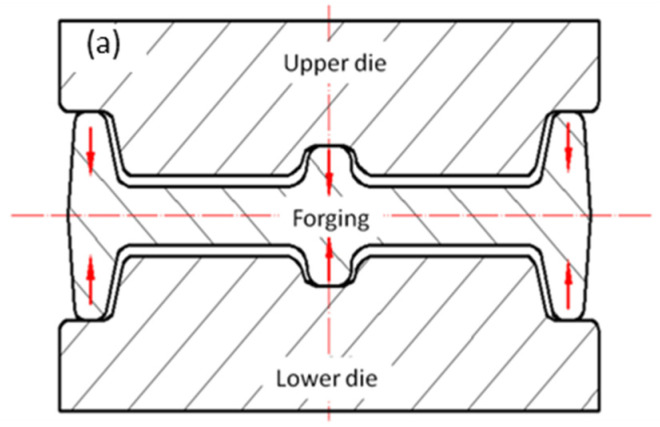
Cold compression of die forging (The red line indicates axis): (**a**) reinforcement bar; (**b**) press bar and press web.

**Figure 3 materials-16-05129-f003:**
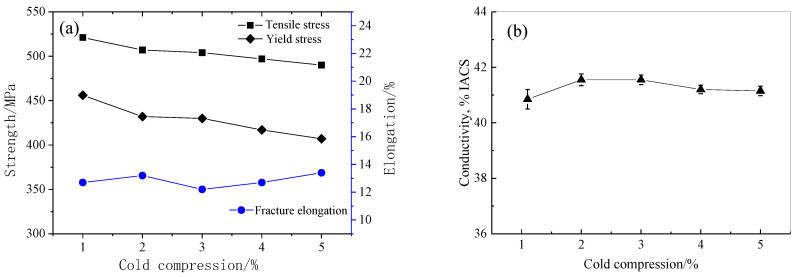
Tensile property and conductivity curves of different cold pressures: (**a**) tensile properties; (**b**) conductivity.

**Figure 4 materials-16-05129-f004:**
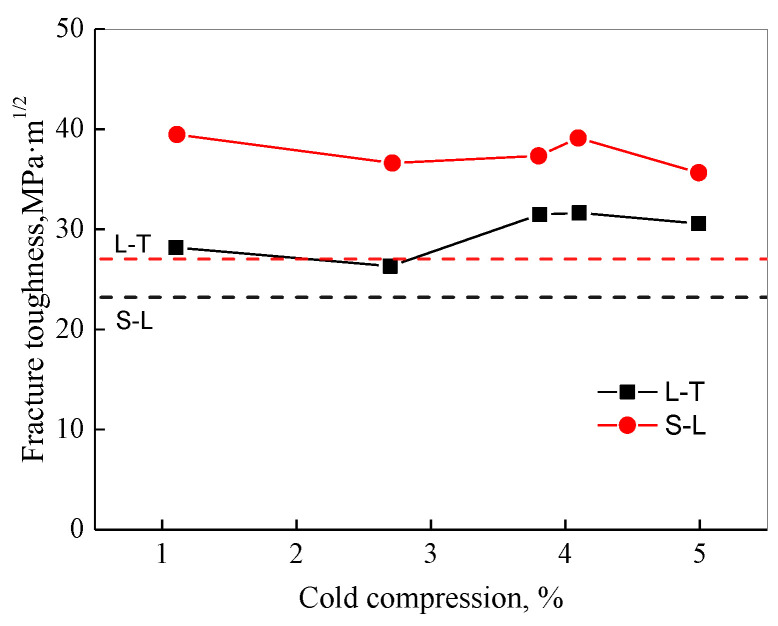
Fracture toughness curves of different cold pressures.

**Figure 5 materials-16-05129-f005:**
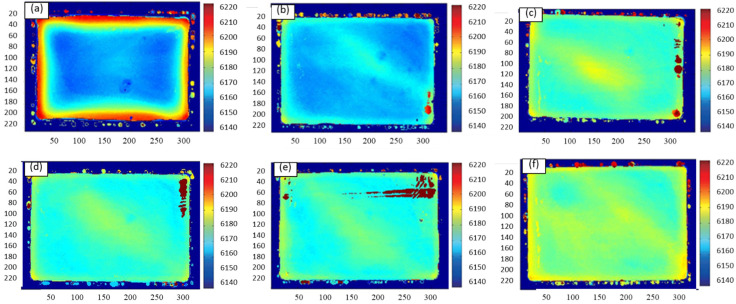
Contour maps of different cold compression percentages: (**a**) uncold pressed; (**b**) 1% cold compression; (**c**) 2% cold compression; (**d**) 3% cold compression; (**e**) 4% cold compression; (**f**) 5 % cold compression.

**Figure 6 materials-16-05129-f006:**
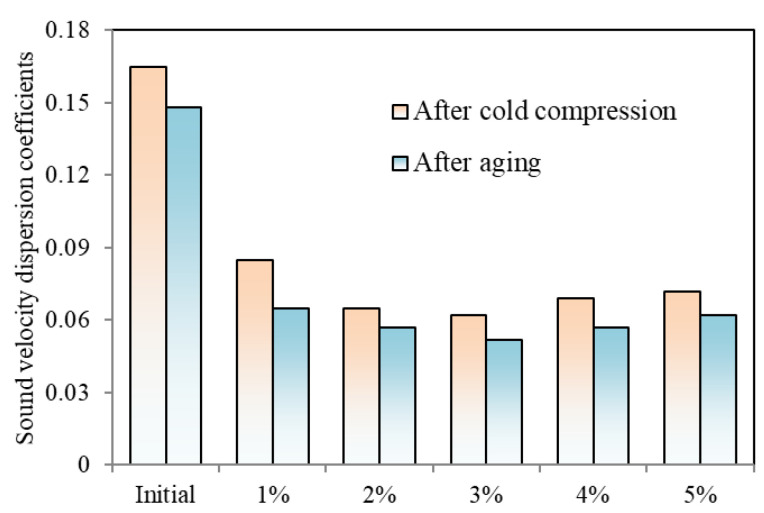
Sound velocity dispersion coefficients.

**Figure 7 materials-16-05129-f007:**
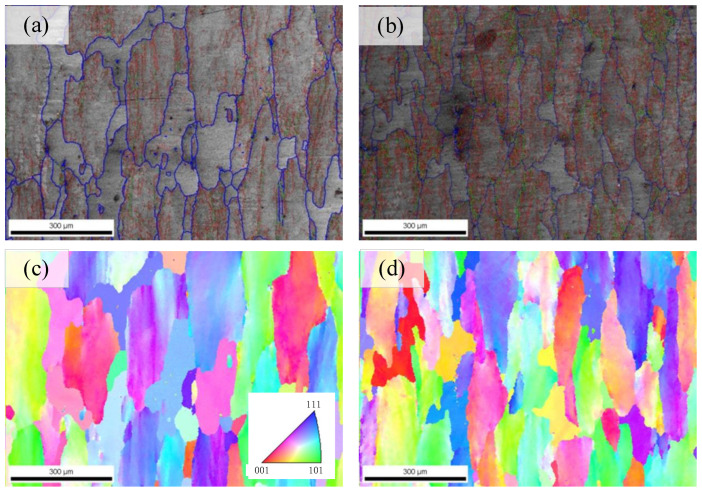
EBSD results of various compression strains: (**a**,**c**) compression strain of 1%, (**b**,**d**) compression strain of 5%.

**Figure 8 materials-16-05129-f008:**
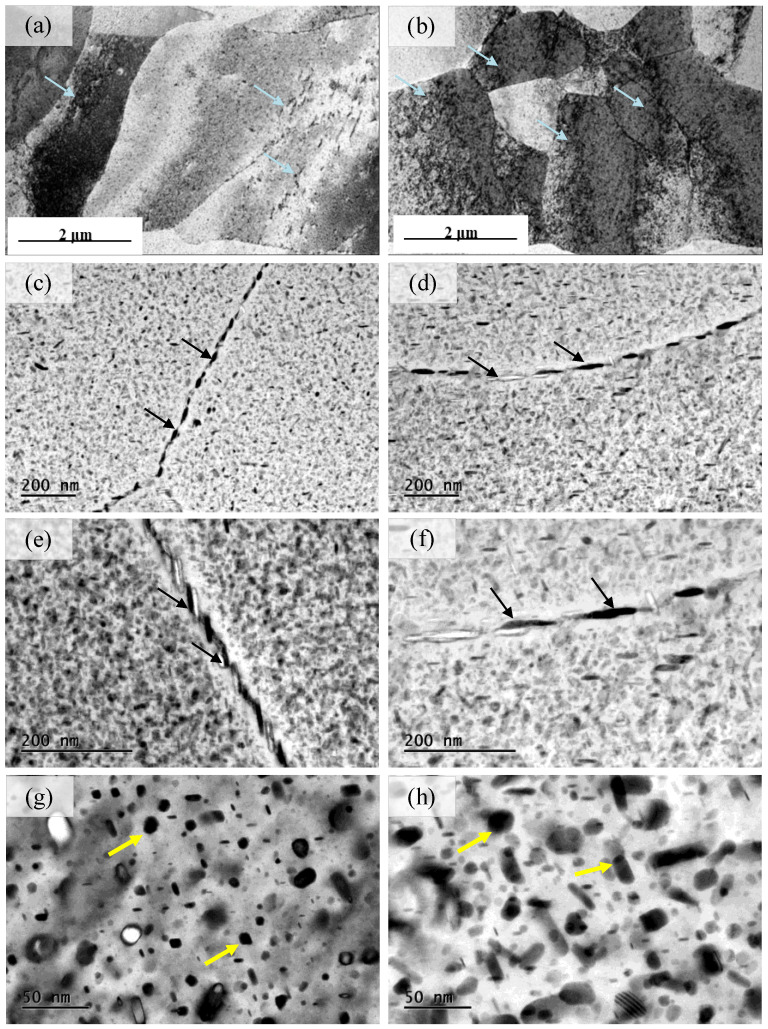
TEM results of various compression strains: (**a**,**c**,**e**,**g**) compression strain of 1%, (**b**,**d**,**f**,**h**) compression strain of 5%, where the dislocation is presented by the blue arrows, the precipitation-phase morphology on the grain boundary are presented by the black arrows, the precipitation phases in the crystal are presented by the yellow arrows.

**Figure 9 materials-16-05129-f009:**
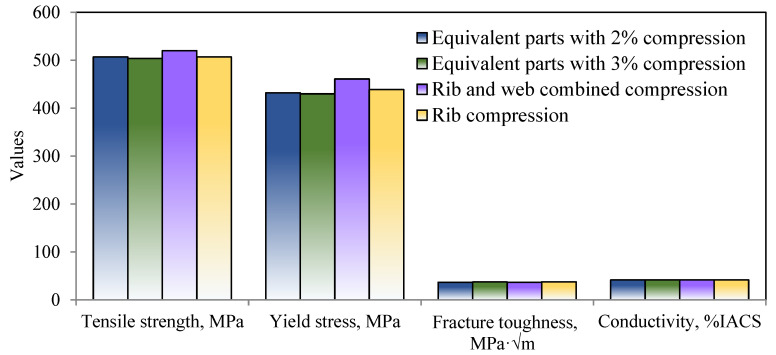
Comparison of mechanical properties between die forging and stocks.

**Figure 10 materials-16-05129-f010:**
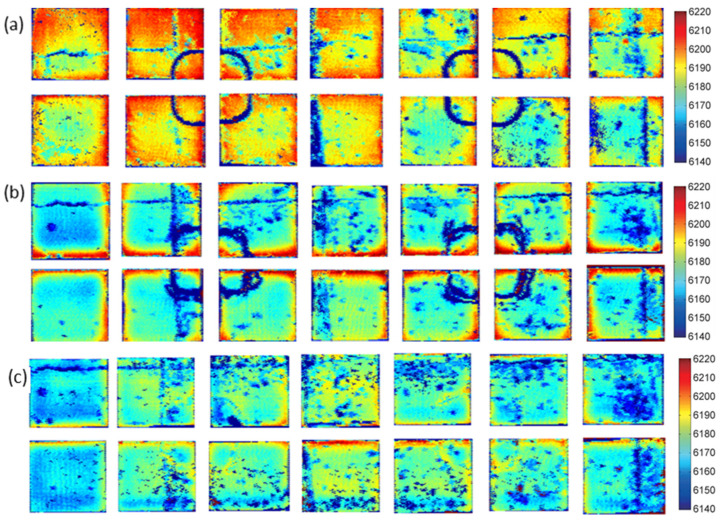
Ultrasonic residual stress test results: (**a**) unpressed samples; (**b**) rib compression; (**c**) rib-and-web combined compression.

**Figure 11 materials-16-05129-f011:**
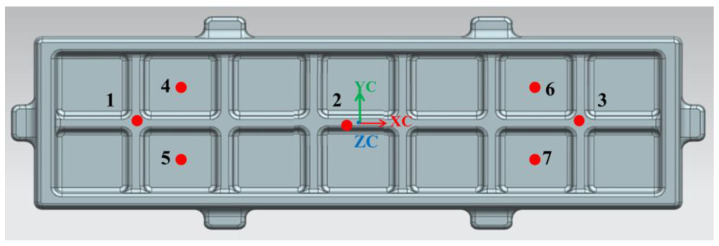
Residual stress detection points. (Numbers are residual stress detection points).

**Figure 12 materials-16-05129-f012:**
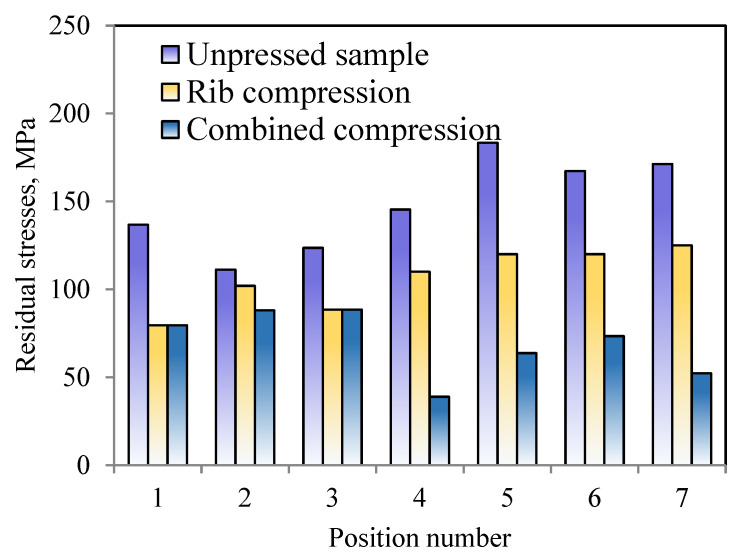
Residual stress comparison results of die forging.

**Figure 13 materials-16-05129-f013:**
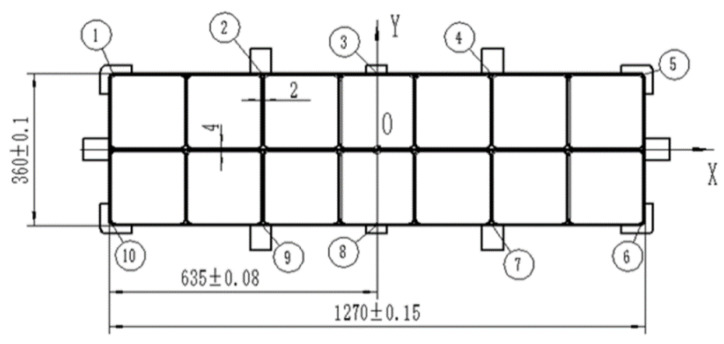
Location measuring point for die forging. (①–⑩ are deformation measurement points).

**Figure 14 materials-16-05129-f014:**
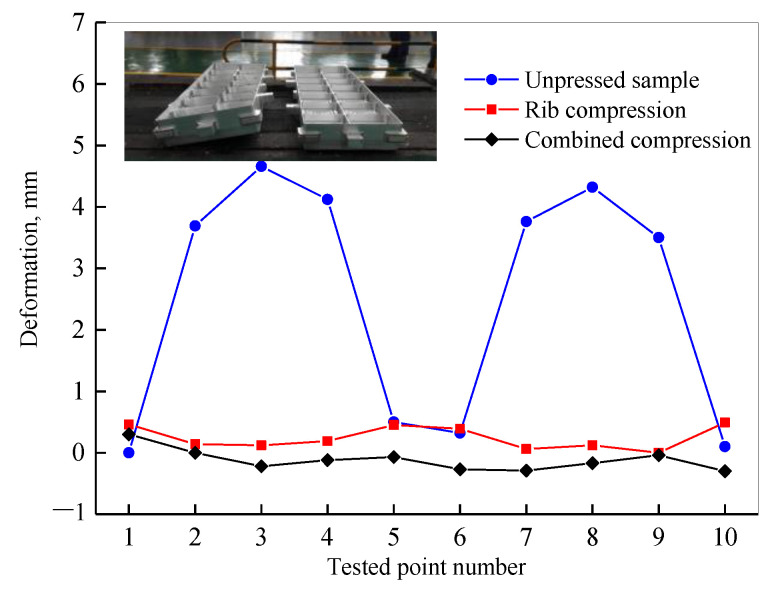
Machining deformation of die forging parts.

**Table 1 materials-16-05129-t001:** The manufacture process parameters.

Process	Parameters
Solution and quenching	477 °C, 60 °C of water
Cold compression	1%, 2%, 3%, 4%, 5%
Aging process	121 °C × 6 h + 177 °C × 8 h

## Data Availability

Not applicable.

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
