# Peer review of "Effect of Cold Pressing Deformation on Microstructure and Residual Stress of 7050 Aluminum Alloy Die Forgings"

_materials, 2023, doi:10.3390/ma16145129_

Round 1
Reviewer 1 Report
I don’t recommend the publication of this paper because of the following reasons.
• Incomplete reference list.
Necessary information is not supplied for [8] to [12] and [18] to [19]. It seems that [7] does not appear in text.
• Ambiguous meaning of “residual stress” used in this paper.
Stress is a tensor and has components. It should be described which component or which part is discussed in this paper.
At Line 119, it is described that “the residual stress is qualitatively obtained according to the change of velocity” by citing [22]. It may be difficult to obtain and read this reference for most of readers of Materials. I feel it is needed to show what’s discussed in [22]. The same thing can be said for [23] at Line 146.
In Fig. 5, experimental results concerning residual stress distribution are presented as a function of position. However, in Fig. 6, the results are summarized as single values excluding the position dependence. This procedure should be explained.
• Errors in Figures.
Four colors are used in Fig. 9 to distinguish four specimens. Although four results using the colors are presented for “Yield stress, MPa (NOT Mps)”, these colors are missing, for example, for the results of “Conductivity, %IACS”. The same thing can be said for Fig. 12.
Figure captions should be modified.
Author Response
When reading the response below, for convenience please note:
- Texts in italic style are the comments from the reviewers.
- Texts in regular style in blue are our responses to the comments.
- Revisions are highlighted using yellow background in the revised manuscript.
Point 1: Incomplete reference list. Necessary information is not supplied for [8] to [12] and [18] to [19]. It seems that [7] does not appear in text.
Response 1: Thanks for the comments. The reference has been revised carefully based on the reviewer’s suggestions.
Reference
[7]Fan N.; Xiong B.; Li Z.; Yan H.; Zhang Y.; Li X.; Wen K. Influence of pre-stretched ratio on surface residual stress of 7055 aluminum alloy thick plate. The Chinese Journal of Nonferrous Metals 2020, 30(2), 301-307.
[8] Pan R.; Zheng J.; Zhang Z.; Lin J. Cold rolling influence on residual stresses evolution in heat-treated AA7xxx T-section panels, Materials and Manufacturing Processes 2019, 34, 431-446.
[9] Zhang Z.; Yang Y.; Li L.; Chen B.; Tai B. Assessment of residual stress of 7050-T7452 aluminum alloy forging using the contour method. Materials Science and Engineering: A 2015, 644, 61-68.
[10] Robinson J.S.; Hossain S.;, Truman C.E.; Paradowska A.M.; Hughes D.J.; Wimpory R.C.; Fox M.E.Residual stress in 7449 aluminium alloy forgings. Materials Science and Engineering: A 2010, 527, 2603-2612.
[11] Gao H.; Wu S.; Wu Q.; Li B.; Gao Z.; Zhang Y.; Mo S. Experimental and simulation investigation on thermal-vibratory stress relief process for 7075 aluminium alloy. Materials & Design 2020, 195,108954.
[12] Liu J.; Du Z.; Su J.; Tang J.; Jiang F.; Fu D.; Teng J.; Zhang H. Effect of quenching residual stress on precipitation behaviour of 7085 aluminium alloy. J. Mater. Sci. Technol. 2023, 132, 154-165.
[16] Wang Z.; SunJ.; Liu L.; Wang R.; Chen W. An analytical model to predict the machining deformation of frame parts caused by residual stress. Journal of Materials Processing Technology 2019, 274, 116282.
[17] Tang J.; Jiang F.; Luo C.; Bo G.; Chen K.; Teng J.; Fu F.; Zhang H. Integrated physically based modeling for the multiple static softening mechanisms following multi-stage hot deformation in Al-Zn-Mg-Cu alloys. International Journal of Plasticity 2020, 134, 102809.
[18] Jiang F.; Zhang H.; Li L.; Chen J. The kinetics of dynamic and static softening during multistage hot deformation of 7150 aluminum alloy. Materials Science and Engineering: A 2012, 552, 269-275.
[19] Jiang F.; Zurob H.S.; Purdy G.R.; Zhang H. Static softening following multistage hot deformation of 7150 aluminum alloy: Experiment and modeling. Materials Science and Engineering: A 2015, 648, 164-177.
Point 2: Ambiguous meaning of “residual stress” used in this paper. Stress is a tensor and has components. It should be described which component or which part is discussed in this paper.
Response 2: Thanks for the comments. I am sorry that there is an ambiguous description of “residual stress”. The blind hole method is commonly applicated to test the surface residual stress which is plane stress state.
Point 3: At Line 119, it is described that “the residual stress is qualitatively obtained according to the change of velocity” by citing [22]. It may be difficult to obtain and read this reference for most of readers of Materials. I feel it is needed to show what’s discussed in [22]. The same thing can be said for [23] at Line 146.
Response 3: Thanks for this very suggestive comment. The descriptions for ultrasonic method and the generation of residual stress are rewritten. And the detailed discussions of reference [22] and [23] which are helpful for this article have been added in section Experimental methods and Residual stress of stocks.
- Experimental methods
Masterscan-380M ultrasonic flaw detector produced by Sonatest is used for ultrasonic testing with a center frequency of 5 MHz and a wafer diameter of Φ 5 inches. The ultrasonic method is a non-destructive analysis method for residual stress based on the self-balancing characteristics of residual stress. During ultrasonic residual stress testing, the sound velocity is changed with material residual stress, and the residual stress is qualitatively obtained according to the change of velocity, where the increase of difference in sound velocity denotes the larger the residual stress [22].
3.2. Residual stress of stocks
Figure 5a shows the cloud of the specimen before cold compression, where the large velocity difference between the surface and core indicates the large residual stress difference in the specimen. During the quenching process, the outside of the material shrinks earlier with the faster cooling rate than the inside of the material attributed to the temperature gradient from the surface to the core. When the temperature of outside is too low to shrink, the inside still has a higher temperature and continues to shrink over time but is restrained by outside material, resulting in the state of tensile stress at core and compressive stress at surface [23].
Point 4: In Fig. 5, experimental results concerning residual stress distribution are presented as a function of position. However, in Fig. 6, the results are summarized as single values excluding the position dependence. This procedure should be explained.
Response 4: Thanks for the comments. Figure 5 plotted a sound velocity map of the block which is obtained by ultrasonic method, where the colors difference represents sound speed variations which are used to imply the stress difference. Figure 6 calculate the dispersion degree of sound velocity to value the sound velocity, which further imply the residual stress variations.
Point 5: Four colors are used in Fig. 9 to distinguish four specimens. Although four results using the colors are presented for “Yield stress, MPa (NOT Mps)”, these colors are missing, for example, for the results of “Conductivity, %IACS”. The same thing can be said for Fig. 12.
Response 5: Thanks for the comments. The errors in figure 9 and 12 have been corrected. We also modified the other figures. Thanks for your kindly revisions.

Reviewer 2 Report
Dear Authors,
Your effort is appreciated, yet the presentation of your research data needs improvement. I encountered serious difficulty in understanding and following your reserach.
First of all the title suggests the influence on microstructure, yet this aspect is vaguely studied (only 4 images).
Also it is unclear to me what do you mean by "compression rate", which makes understanding more difficult.
Your introduction is mainly a list of the work of other authors and vaguely related to yours. I suggest explaining in more depth the concepts you used.
The experimental program is confusing and the standards used need to be related to ASTM or ISO standards ( personally I am unaware of the contents of the standards you mentioned). I reccommend rewriting the paper and inluding a section stating the materials and methods.
Something that I failed to understand is why the mechanical characteristics are decreasing as you state that dislocation density increases and phase precipitation occurs.
Your results have potential but a more comprehensive presentation is required. I strongly encourage you in rewriting the paper.
Also several references included lack authors, journals or any identification.
My best regards.
Serious English grammar and style editing is required.
Author Response
When reading the response below, for convenience please note:
- Texts in italic style are the comments from the reviewers.
- Texts in regular style in blue are our responses to the comments.
- Revisions are highlighted using yellow background in the revised manuscript.
We are thankful to the reviewer for the encouraging and helpful comments to improve the manuscript. We have considered the suggestions made by the reviewer and have accordingly modified the manuscript. We are grateful that the reviewer finds our work interesting and have potential for future applications. The specific comments are answered as follows:
Point 1: First of all, the title suggests the influence on microstructure, yet this aspect is vaguely studied (only 4 images).
Response 1: Thanks for the comments. The microstructure analysis has been added including EBSD and TEM results following reviewer’s suggestion, in order to present a better illustration of microstructure evolution with compression strain. The section “3.3. Microstructure morphology” has been written.
3.3. Microstructure morphology
Figure 7 shows EBSD results of samples with various compression strain. It is shown that the samples after cold compression present a recrystallized crystal morphology. The recrystallized grain proportion of sample with compression strain of 1% is significantly higher than that of 5%.
Figure 8 plots the TEM results of samples with various compression strains. Figures 8 a and b illustrate dislocation variation between different compression strain. The sample with 5% compression strain has a large number of dislocation density and a dislocation cell structure is formed due to dislocation entanglement while the dislocation density of sample with 1% compression strain is little. Figures 8 c-f present the precipitation phase morphology of grain boundary of different material. The grain boundary precipitation phase for 1% strain sample is continuously distributed and its grain size is 50 μm. However, the precipitated phase spacing is increased as the strain increasing from 1% to 5% and showing intermittent distribution, and the grain size is about 100 μm. Figures 8 g and h show the morphological distribution of the precipitated phase in the crystal. The precipitated phase in the crystal of 1% strain sample is smaller than that of 5%. In summary, the increase in cold compression strain increases the dislocation density and induces a larger precipitated phase, where the former increases the material strength while the latter has an opposite effect. As the cold compression strain increases from 1% to 5%, the tensile strength is reduced by 30MPa and the yield strength is decreased by 50MPa, indicating that precipitation plays a greater role in the material properties.
Point 2: Also it is unclear to me what do you mean by "compression rate", which makes understanding more difficult.
Response 2: Thanks for the comments. The compression rate is the ratio of the compression deformation amount to the forging stock thickness, which also is the engineering strain during the compression deformation. We are sorry the incomprehensible description of compression variations, and it could be easier to understand with a description of compressed strain. Thereby, the “compression rate” was modified by “compressive strain” in this article.
Point 3: Your introduction is mainly a list of the work of other authors and vaguely related to yours. I suggest explaining in more depth the concepts you used.
Response 3: Thanks for the comments. This paper investigated the cold compression effect on material properties and microstructure in addition to analysis of residual stress distribution. It is found that the comprehensive properties of materials are not optimized with the increase of compressive strain. Thanks again for your suggestion, and we modified the description of Introduction.
Introduction
Tang et al. [17] proposed an integrated physically based modeling to predict the mechanical properties in hot forging of aluminum alloy. Jiang et al. [18,19] revealed the kinetics of dynamic and static softening during forging process. The researchers have analyzed the influence of cold compression on residual stress through experiments and simulations. However, the evolution of microstructure and mechanical properties during cold compression is rarely reported, resulting in an obstacle of material property control of complex parts during residual stress reduction.
Point 4: The experimental program is confusing and the standards used need to be related to ASTM or ISO standards ( personally I am unaware of the contents of the standards you mentioned). I recommend rewriting the paper and including a section stating the materials and methods.
Response 4: Thanks for the comments. According to reviewer’s suggestion, the experimental methods section has been rewritten to make it easier for readers to understand. Meanwhile, the standards have been modified by the ASTM standard.
- Experimental methods
The forging material is made by 7050 aluminum alloy, which main chemical components are Zn (6.27 wt. %), Cu (2.03 wt. %), Mg (1.96 wt. %) and Zr (0.10 wt. %). In this work, an equivalent sample is selected to analyze the formation mechanism during cold compression, and a large-scale rib-structural forging is studied for industrial application of die forge. The size of forging stock of an equivalent part is 200 mm×150 mm×140 mm, while die forging stock is designed as shown in Figure 1 and have a size of 1316 mm×406 mm×140 mm. The residual stresses are caused by uneven cold shrinkage inside and outside during quenching. Therefore, all stocks are solid solution treated at 477 °C for 210min and are quenching in water at room temperature. Then, cold compression is applied for the samples, where compression strains of equivalent sample are 1%, 2%, 3%, 4%, and 5%, and those of die forging stock are optimized 2~3%. For the die forging stock, cold compression is carried out by special cold compression mold, and according to the structural characteristics of die forgings, cold compression is applied to eliminate residual stress by pressing rib and web respectively, as shown in Figure 2. For the process integrity of the forging production, a two-stage aging process is applied where the samples are treated at 121 °C for 6h, and heated to 177°C and hold for 8h.
Three samples of room temperature tension are repeated for results accuracy which have a gauge length of 32 mm and a diameter of 6 mm and their size is following standard of ASTM B557. The fracture toughness is tested according to standard of ASTM E399 and the samples are cut from parts along L-T and S-L directions with size of 20 mm×46 mm×50 mm. The conductivity results are tested on the surface of tensile samples and repeated for three times following the standard of ASTM E1004.
Point 5: Something that I failed to understand is why the mechanical characteristics are decreasing as you state that dislocation density increases and phase precipitation occurs.
Response 5: Thanks for the comments. The increase in cold compression strain increases the dislocation density and induces a larger precipitated phase. The increase of dislocation density increases the material strength while the increase of precipitated phase decreases the material strength. As the cold compression strain increases from 1% to 5%, the tensile strength is reduced by 30MPa and the yield strength is decreased by 50MPa, indicating that precipitation plays a greater role in the material properties. The effect of microstructure on mechanical properties has been added in section 3.3. Microstructure morphology.
Point 6: Your results have potential but a more comprehensive presentation is required. I strongly encourage you in rewriting the paper.
Response 6: Thanks for the comments. The illustration of microstructure evolution with compression strain has been written.
Point 7: Also several references included lack authors, journals or any identification.
Response 7: Thanks for the comments. The reference has been revised carefully based on the reviewer’s suggestions.
Reference
[7]Fan N.; Xiong B.; Li Z.; Yan H.; Zhang Y.; Li X.; Wen K. Influence of pre-stretched ratio on surface residual stress of 7055 aluminum alloy thick plate. The Chinese Journal of Nonferrous Metals 2020, 30(2), 301-307.
[8] Pan R.; Zheng J.; Zhang Z.; Lin J. Cold rolling influence on residual stresses evolution in heat-treated AA7xxx T-section panels, Materials and Manufacturing Processes 2019, 34, 431-446.
[9] Zhang Z.; Yang Y.; Li L.; Chen B.; Tai B. Assessment of residual stress of 7050-T7452 aluminum alloy forging using the contour method. Materials Science and Engineering: A 2015, 644, 61-68.
[10] Robinson J.S.; Hossain S.;, Truman C.E.; Paradowska A.M.; Hughes D.J.; Wimpory R.C.; Fox M.E.Residual stress in 7449 aluminium alloy forgings. Materials Science and Engineering: A 2010, 527, 2603-2612.
[11] Gao H.; Wu S.; Wu Q.; Li B.; Gao Z.; Zhang Y.; Mo S. Experimental and simulation investigation on thermal-vibratory stress relief process for 7075 aluminium alloy. Materials & Design 2020, 195,108954.
[12] Liu J.; Du Z.; Su J.; Tang J.; Jiang F.; Fu D.; Teng J.; Zhang H. Effect of quenching residual stress on precipitation behaviour of 7085 aluminium alloy. J. Mater. Sci. Technol. 2023, 132, 154-165.
[16] Wang Z.; SunJ.; Liu L.; Wang R.; Chen W. An analytical model to predict the machining deformation of frame parts caused by residual stress. Journal of Materials Processing Technology 2019, 274, 116282.
[17] Tang J.; Jiang F.; Luo C.; Bo G.; Chen K.; Teng J.; Fu F.; Zhang H. Integrated physically based modeling for the multiple static softening mechanisms following multi-stage hot deformation in Al-Zn-Mg-Cu alloys. International Journal of Plasticity 2020, 134, 102809.
[18] Jiang F.; Zhang H.; Li L.; Chen J. The kinetics of dynamic and static softening during multistage hot deformation of 7150 aluminum alloy. Materials Science and Engineering: A 2012, 552, 269-275.
[19] Jiang F.; Zurob H.S.; Purdy G.R.; Zhang H. Static softening following multistage hot deformation of 7150 aluminum alloy: Experiment and modeling. Materials Science and Engineering: A 2015, 648, 164-177.
Point 8: Comments on the Quality of English Language Serious English grammar and style editing is required.
Response 8: Thanks for the comments. The manuscript has been revised based on the reviewer’s suggestions. The grammatical mistakes have been corrected. The whole manuscript is examined by a professional language proofreader.

Reviewer 3 Report
The paper presents experimental research in the field of aluminum alloy (7050 Aluminum Alloy) especially regarding the cold plastic deformation. Five compression rates were applied on a part with dimensions equal to 200 mm×150 mm×140. The methodology is well described and the results show the differences between the samples function the compression rate and heat treatment.
Please take into account the followings editing changes:
- Fig. 2 please make the dimensions more clearly
- Fig. 3 is not so clear. Try to use the same format like in figure 14 for all the graphs
- Fig. 10, the scale in the right part is unclear
Author Response
When reading the response below, for convenience please note:
- Texts in italic style are the comments from the reviewers.
- Texts in regular style in blue are our responses to the comments.
- Revisions are highlighted using yellow background in the revised manuscript.
The paper presents experimental research in the field of aluminum alloy (7050 Aluminum Alloy) especially regarding the cold plastic deformation. Five compression rates were applied on a part with dimensions equal to 200 mm×150 mm×140. The methodology is well described and the results show the differences between the samples function the compression rate and heat treatment.
We are thankful to the reviewer for the helpful comments to improve the manuscript. According to the reviewer’s comments, we have revised manuscript. We hope that our revision, based on the input of the reviewer’s comments, has resolved this concern.
Point 1: Please take into account the followings editing changes: Fig. 2 please make the dimensions more clearly
Response 1: Thanks for the comments. I am sorry for unclear illustration of figure 2. The figure 2 has been replotted to make it clear according to reviewer’s suggestion. Thanks for your kindly comments again.
Point 2: Fig. 3 is not so clear. Try to use the same format like in figure 14 for all the graphs
Response 2: Thank you for your rigorous and kindly comments. Figure 3 has been modified according to your suggestion.
Point 3: Fig. 10, the scale in the right part is unclear
Response 3: Thanks for the comments. The color bar of figure 10 has been modified for more clear illustration. Thanks for your meticulous review and kindly suggestions.

Reviewer 4 Report
This manuscript presents results about the impact of cold compression rate on the mechanical properties, microstructure and residual stress of 7075 aluminum alloy to be used for die forging. Although this is an interesting topic and has industrial applications, the reviewer believes that the manuscript needs to be reworked. Some sections require more explanations and details. Therefore, the reviewer suggests that the manuscript requires Major Revisions.
1. Introduction
1) Previous works have been mentioned and reported nicely. However, the novelty of this study has not been described properly. Please specify what is novel about this work at end of Introduction section.
2. Experimental Methods
1) An actual image of each part (i.e. die forging, press bar, and press web) in addition to their dimension would help readers to imagine the geometry of the parts properly. Please consider including some actual images.
2.1 Results of billets
1) There is a duplicate for section 2. (2. Experimental Methods and 2. Results of billets)
2) It is not clear why the conductivity was measured. Given the numbers are so close to one another, how can one make sure about the accuracy of results? Are the slight variations related to movement of dislocations and microstructural evolution of the parts?
3) The numbers on graphs (Figs. 3-5) are too small. Please revise.
4) The study of the microstructure requires more detail. The images are not of good quality and not much can be seen from them. Please consider replacing them with high-quality images.
5) Images taken by optical and scanning electron microscopes can be used to show the microstructural features nicely.
6) Please clearly show the special features such as precipitated phases on the figure with arrows and text boxes.
7) The scale bar and magnification of TEM images (Figs. 7 and 8) cannot be seen. Please revise.
3. Results of Die Forging
1) Quality of Fig. 10 requires special attention. Please consider enlarging the images and numbers so the residual stress values can be seen.
2) The actual images of parts after forging would be informative. For example, Fig. 14 shows machining deformation, but it is not clear where they were measured from. The warping of the actual samples are not shown. Please include some images.
The manuscript is written well, but it can be improved by being proofread by a native English speaker. As an example, it is written in the abstract section that "The results shield the light on the industrial application".
Author Response
When reading the response below, for convenience please note:
- Texts in italic style are the comments from the reviewers.
- Texts in regular style in blue are our responses to the comments.
- Revisions are highlighted using yellow background in the revised manuscript.
The paper presents experimental research in the field of aluminum alloy (7050 Aluminum Alloy) especially regarding the cold plastic deformation. Five compression rates were applied on a part with dimensions equal to 200 mm×150 mm×140. The methodology is well described and the results show the differences between the samples function the compression rate and heat treatment.
We are thankful to the reviewer for the helpful comments to improve the manuscript. According to the reviewer’s comments, we have revised manuscript. We hope that our revision, based on the input of the reviewer’s comments, has resolved this concern.
Point 1: 1. Introduction: 1) Previous works have been mentioned and reported nicely. However, the novelty of this study has not been described properly. Please specify what is novel about this work at end of Introduction section.
Response 1: Thanks for the comments. Previous works are mainly focused on the residual stress variation during cold compression. However, the material properties are the key quality for forge products. Therefore, this work investigated the influence of cold compression parameter on mechanical properties and microstructure evolution in addition to residual stress. The Introduction has been modified according to reviewer’s suggestions.
- Introduction
Tang et al. [17] proposed an integrated physically based modeling to predict the mechanical properties in hot forging of aluminum alloy. Jiang et al. [18,19] revealed the kinetics of dynamic and static softening during forging process. The researchers have analyzed the influence of cold compression on residual stress through experiments and simulations. However, the evolution of microstructure and mechanical properties during cold compression is rarely reported, resulting in an obstacle of material property control of complex parts during residual stress reduction.
The limited investigations of interaction effect between residual stress and mechanical properties affects the quality of complex forging products during cold compression process of complex forging products. In this study, the cold compression is experimentally investigated for integrated control of forging mechanical properties and residual stress based on an equivalent sample and a large-scale rib-structural forging.
Point 2: 2. Experimental Methods:1) An actual image of each part (i.e. die forging, press bar, and press web) in addition to their dimension would help readers to imagine the geometry of the parts properly. Please consider including some actual images.
Response 2: Thanks for the comments. The engineering drawings of the die forging sample while it could be difficult to understand. Therefore, the actual image of die forging productions has been added for an intuitive instruction.
Point 3: 2.1 Results of billets: 1) There is a duplicate for section 2. (2. Experimental Methods and 2. Results of billets)
Response 3: Thanks for the comments. The section number has been modified. The section “2. Results of billets” has been replaced by “3. Results of tocks”.
Point 4: 2.1 Results of billets: 2) It is not clear why the conductivity was measured. Given the numbers are so close to one another, how can one make sure about the accuracy of results? Are the slight variations related to movement of dislocations and microstructural evolution of the parts?
Response 4: Thanks for the comments. Conductivity measurement is conventional test for aluminum alloy to evaluate the aging process meeting the requirements. In this work, the conductivities of various compression strain are closed to each other due to the same aging treatment parameters. The analysis of conductivity has been modified in section 3.1.
3.1. Mechanical property of stocks
The elongation of forgings gradually increases. With the increase of cold pressing capacity, the conductivity of forgings changes little implying the aging process meeting the requirements. Figure 4 shows the fracture toughness test results. The fracture toughness changes little with the increase of cold compression, and the change value is within 5 MPa·m1/2.
Point 5: 2.1 Results of billets: 3) The numbers on graphs (Figs. 3-5) are too small. Please revise.
Response 5: Thanks for the comments. The figures 3-5 have been replotted following the reviewer’s suggestions for a better presentation of results.
Point 6: 2.1 Results of billets: 4) The study of the microstructure requires more detail. The images are not of good quality and not much can be seen from them. Please consider replacing them with high-quality images.
Response 6: Thanks for the comments. The microstructure evolution including EBSD and TEM results has been added for more detail analysis. The figures 7 and 8 has been replaced by high-quality images according to reviewer’s kindly suggestion. The section “3.3. Microstructure morphology” has been written.
3.3. Microstructure morphology
Figure 7 shows EBSD results of samples with various compression strain. It is shown that the samples after cold compression present a recrystallized crystal morphology. The recrystallized grain proportion of sample with compression strain of 1% is significantly higher than that of 5%.
Figure 8 plots the TEM results of samples with various compression strains. Figures 8 a and b illustrate dislocation variation between different compression strain. The sample with 5% compression strain has a large number of dislocation density and a dislocation cell structure is formed due to dislocation entanglement while the dislocation density of sample with 1% compression strain is little. Figures 8 c-f present the precipitation phase morphology of grain boundary of different material. The grain boundary precipitation phase for 1% strain sample is continuously distributed and its grain size is 50 μm. However, the precipitated phase spacing is increased as the strain increasing from 1% to 5% and showing intermittent distribution, and the grain size is about 100 μm. Figures 8 g and h show the morphological distribution of the precipitated phase in the crystal. The precipitated phase in the crystal of 1% strain sample is smaller than that of 5%. In summary, the increase in cold compression strain increases the dislocation density and induces a larger precipitated phase, where the former increases the material strength while the latter has an opposite effect. As the cold compression strain increases from 1% to 5%, the tensile strength is reduced by 30MPa and the yield strength is decreased by 50MPa, indicating that precipitation plays a greater role in the material properties.
Point 7: 2.1 Results of billets: 5) Images taken by optical and scanning electron microscopes can be used to show the microstructural features nicely.
Response 7: Thanks for the comments. The SEM/EBSD results of samples with various compression strain has been added in Figure 7.
Point 8: 2.1 Results of billets: 6) Please clearly show the special features such as precipitated phases on the figure with arrows and text boxes.
Response 8: Thanks for the comments. The precipitated phases has been illustrated using the arrows in figure 8. Thanks for your suggestive comments again.
Point 9: 2.1 Results of billets: 7) The scale bar and magnification of TEM images (Figs. 7 and 8) cannot be seen. Please revise.
Response 9: Thanks for the comments. The figures 7 and 8 has been replaced by high-quality images. Thank you again for your meticulous review.
Point 10: 3. Results of Die Forging: 1) Quality of Fig. 10 requires special attention. Please consider enlarging the images and numbers so the residual stress values can be seen.
Response 10: Thanks for the comments. The color bar of figure 10 has been modified for more clear illustration. Thanks for your meticulous review and kindly suggestions.
Point 11: 3. Results of Die Forging: 2) The actual images of parts after forging would be informative. For example, Fig. 14 shows machining deformation, but it is not clear where they were measured from. The warping of the actual samples are not shown. Please include some images.
Response 11: Thanks for the comments. The actual images of parts have been added in Figure 14 to illustrate the deformation after machining process. It is shown that the ribs are deviated from the designed shape implying a serious geometrical errors induced by residual stress after machining.
Point 12: The manuscript is written well, but it can be improved by being proofread by a native English speaker. As an example, it is written in the abstract section that "The results shield the light on the industrial application".
Response 12: Thanks for the comments. The manuscript has been revised based on the reviewer’s suggestions. The grammatical mistakes have been corrected. The whole manuscript is examined by a professional language proofreader.

Round 2
Reviewer 1 Report
Now I recommend the publication of this paper.
Author Response
Thank you for taking your time to review our manuscript. We would appreciate your valuable advice and great help.
Reviewer 2 Report
Dear Authors,
I appreciate the efforts in improving your research. Several minor editing errors are still present:
line 39 and 42: the reference to Robinson appears for [8] and [10], but they do not correspond to the list at the end. Adress this issue.
in line 69, I recommend that immediately after Zhai et al. you include the reference, Zhai et al. [21]
line 88: the forging material is 7050 aluminum alloy and the alloying elements are Zn ...
line 125: diameter of 5 inches needs to be expressed in milimeter
line 138-139: rephrase
line 190: of grain boundary of different material. - rephrase, precipitation phase on the grain boundary or another way
line 191: grain size - please use average grain size instead
line 194: distribution of the precipitated phase in the crystal. - rephrase
line 197: larger precipitated phase - rephrase, the dimensions of the precipitated phase are larger, it can be interpreted that the phase proportion is larger
In fig. 8 the black and yellow arrows have no mention in text. Make a reference in text where the discussion is appropriate for them.
line 208: rephrase subtitle 4.1
Line 254: The machining experiment is continued - rephrase
and geometric errors are measured - rephrase, the errors are not measured directly
line 257: largest mistake - rephrase
line 262: check reported values, maximum error is 0.3mm for the combined compression method and 0.3mm for the rip compression.
Please adress these issues.
My best regards.
Minor English corrections are required, please check the comments.
Author Response
Response to Reviewer Comments
When reading the response below, for convenience please note:
- Texts in italic style are the comments from the reviewers.
- Texts in regular style in blue are our responses to the comments.
- Revisions are highlighted using yellow background in the revised manuscript.
I appreciate the efforts in improving your research. Several minor editing errors are still present.
We are thankful to the reviewer for the helpful comments to improve the manuscript. We have considered the suggestions made by the reviewer and have accordingly modified the manuscript.The specific comments are answered as follows:
Point①:line 39 and 42: the reference to Robinson appears for [8] and [10], but they do not correspond to the list at the end. Address this issue.
Point②:in line 69, I recommend that immediately after Zhai et al. you include the reference, Zhai et al. [21]
Response①~②:
- Introduction
Pan et al. [8] studied the influence of quench sensitivity on residual stress.
Zhai et al. [21] predicted the residual stress distribution of long rib free forgings with dimensions of 1500 mm×500 mm ×200 mm after quenching using thermodynamic calculation software and plastic forming software, and found 3% cold pressing deformation and 200 mm feed amount can eliminate the quenching residual stress.
Point③: line 88: the forging material is 7050 aluminum alloy and the alloying elements are Zn ...
Response③:
- Experimental methods
The forging material is 7050 aluminum alloy and the alloying elements are Zn (6.27 wt. %), Cu (2.03 wt. %), Mg (1.96 wt. %) and Zr (0.10 wt. %).
Point④:line 125: diameter of 5 inches needs to be expressed in milimeter
Response④:
The residual stresses of the equivalent part and rib-structural forgings is detected by ultrasonic method and blind hole method. Masterscan-380M ultrasonic flaw detector produced by Sonatest is used for ultrasonic testing with a center frequency of 5 MHz and a wafer diameter of Φ 12.7 mm.
Point⑤:line 138-139: rephrase
Response⑤:
- Results of stocks
With the increase of cold compression strain, the conductivity of forgings changes little implying the aging process meets the requirements. Figure 4 shows the fracture toughness test results.
Point⑥: line 190: of grain boundary of different material. - rephrase, precipitation phase on the grain boundary or another way
Point⑦: line 191: grain size - please use average grain size instead
Point⑧: line 194: distribution of the precipitated phase in the crystal. - rephrase
Point⑨: line 197: larger precipitated phase - rephrase, the dimensions of the precipitated phase are larger, it can be interpreted that the phase proportion is larger
Point⑩: In fig. 8 the black and yellow arrows have no mention in text. Make a reference in
Response⑥~⑩:
Figure 8 plots the TEM results of samples with various compression strains. Figures 8 a and b illustrate dislocation variation between different compression strain, where the dislocation are presented by yellow arrow. The sample with 5% compression strain has a large number of dislocation density and a dislocation cell structure is formed due to dislocation entanglement while the dislocation density of sample with 1% compression strain is little. Figures 8 c-f present the precipitation phase morphology on the grain boundary of different material, where the precipitation phases are presented by black arrow. The grain boundary precipitation phase for 1% strain sample is continuously distributed and its grain size is 50 μm. However, the precipitated phase spacing is increased as the strain increasing from 1% to 5% and showing intermittent distribution, and the average grain size is about 100 μm. Figures 8 g and h show the morphological distribution of the precipitated phase in the crystal, where the precipitation phases are presented by yellow arrow. The precipitated phase in the crystal of 1% strain sample is smaller than that of 5%. In summary, the increase in cold compression strain increases the dislocation density and induces a larger proportion of precipitated phase, where the former increases the material strength while the latter has an opposite effect.
Point⑪: line 208: rephrase subtitle 4.1
Response⑪:
- Results of die forging
4.1. Mechanical properties with various compression methods
Point⑫: Line 254: The machining experiment is continued – rephrase
and geometric errors are measured - rephrase, the errors are not measured directly
Point⑬:line 257: largest mistake - rephrase
Point⑭:line 262: check reported values, maximum error is 0.3mm for the combined compression method and 0.3mm for the rip compression.
Response⑫~⑭:
4.3. Die forging machining result
The machining experiment is further carried out and ten geometric deformation measurement points are shown in Figure 13. The geometric deformations of various positions are shown in Figure 14, where the unpressed sample exhibits a significant inaccuracy in the middle of sections with positions of 2~4 and 7~9. The position 3 error is the spot with the largest error, measuring 4.7 mm. By using the cold compression method, the errors are reduced, indicating that the machining deformation is suppressed as residual stress decreases. Due to the lower residual stress, the combined compression technique's errors are substantially lower than those of the rib compression method; the combined compression method's maximum error is 0.3 mm, whereas the rib compression method's is 0.49 mm. According to the aforementioned findings, rib and web compression with a compression strain of 2~3% is helpful in reducing residual stress and improving geometric precision while maintaining the material's mechanical properties.
Reviewer 4 Report
The authors have responded to all the questions and have implemented all the comments. The quality of the manuscript has improved nicely. Thus, the reviewer believes that it is ready for publication.
Author Response

(The authors gave the same response as above.)
